# Associations between Depression and Self-Reported COVID-19 Symptoms among Adults: Results from Two Population-Based Seroprevalence Studies in Switzerland

**DOI:** 10.3390/ijerph192416696

**Published:** 2022-12-12

**Authors:** Giovanni Piumatti, Rebecca Amati, Aude Richard, Hélène Baysson, Marianna Purgato, Idris Guessous, Silvia Stringhini, Emiliano Albanese

**Affiliations:** 1Fondazione Agnelli, 10125 Turin, Italy; 2Institute of Public Health, Faculty of BioMedicine, Università della Svizzera Italiana, 6900 Lugano, Switzerland; 3Institute of Global Health, University of Geneva, 1202 Geneva, Switzerland; 4Division of Primary Care, Geneva University Hospitals, 1206 Geneva, Switzerland; 5Department of Health and Community Medicine, Faculty of Medicine, University of Geneva, 1211 Geneva, Switzerland; 6Section of Psychiatry, WHO Collaborating Centre for Research and Training in Mental Health and Service Evaluation, Department of Neuroscience, Biomedicine, and Movement Sciences, University of Verona, 37134 Verona, Italy; 7Cochrane Global Mental Health, University of Verona, 37129 Verona, Italy

**Keywords:** COVID-19, depressive symptoms, seropositivity, population-based observational study

## Abstract

(1) Mental health may modulate the perceived risk of SARS-CoV-2 infection. However, it is unclear how psychological symptoms may distort symptom perception of COVID-19 and SARS-CoV-2 infection. We assessed whether depressive symptoms predicted self-reported COVID-19 symptoms, independently of serologically confirmed SARS-CoV-2 infection. (2) Participants (aged 20–64) in the Geneva (N = 576) and Ticino (N = 581) Swiss regions completed the Patient Health Questionnaire before being tested for anti-SARS-CoV-2 IgG antibodies and recalled COVID-19-compatible symptoms on two occasions: April–July 2020 (baseline), and January–February 2021 (follow-up). We estimated prevalence ratios for COVID-19 symptoms by depression scores in interaction with serological status. (3) At baseline, in Geneva, higher depression predicted higher probability of reporting systemic, upper airways, and gastro-intestinal symptoms, and fever and/or cough; in Ticino, higher depression predicted systemic, upper airways, and gastro-intestinal symptoms, fever and/or cough, dyspnea, and headache. At follow-up, in Geneva, higher depression predicted higher probability of reporting systemic symptoms and dyspnea; in Ticino, higher depression predicted higher probability of reporting systemic and upper airways symptoms, dyspnea and headache (all *p* values < 0.05). (4) We found positive associations between depressive symptoms and COVID-19-compatible symptoms, independently of seropositivity. Mental wellbeing has relevant public health implications because it modulates self-reported infection symptoms that inform testing, self-medication, and containment measures, including quarantine and isolation.

## 1. Introduction

The COVID-19 pandemic has increased the risk of psychological distress and mental disorders in the general population [1,2]. The mental health of individuals may have disproportionally worsened during the pandemic in those with pre-existing, pre-pandemic mental disorders [3,4], and poor mental health may be associated with a higher perceived risk of infection at the population level [5,6,7].

Depression can cause hyperawareness or altered body perception and is associated with somatic symptoms and medical illness catastrophizing [8,9]. According to both the tenth revision of the International Statistical Classification of Diseases and Related Health Problems (ICD-10) [10], and the fifth edition of the Diagnostic and Statistical Manual of Mental Disorders (DSM-5) [11], hypochondriacal symptoms are considered secondary, accompanying symptoms of major depression. Depressive symptoms may thus complexify COVID-19 detection and diagnosis. However, no study has yet investigated whether and the extent to which depressive symptoms modify or distort perceptions of COVID-19 symptomatology in people with and without a serologic confirmed SARS-CoV-2 infection.

In the present study, we aimed to assess whether higher depressive symptoms increased the likelihood of reporting COVID-19-compatible symptoms during the first and the second pandemic waves in Switzerland, accounting for serological evidence of previous SARS-CoV-2 infection. We used data from two population-based seroprevalence studies with similar designs conducted in two Swiss regions (Geneva and Ticino) [12,13].

## 2. Materials and Methods

### 2.1. Study Design and Participants

An individual participant dataset with repeated observations on a sample of 1157 adult participants was used. The dataset pooled data from two studies with a similar design, namely the SEROCoV-POP study in Geneva, and the Corona Immunitas study in Ticino, Switzerland. Full details about sampling, recruitment and data collection procedures for both studies have been previously described [12,13]. Consistent with the two original study designs, we focus on adults aged 20 to 74. Briefly, the SEROCoV-POP study is a population-based observational seroprevalence study including former participants of the Bus Santé study [14], a repeated cross-sectional population-based health survey in the canton of Geneva. Every year, about 1000 Bus Santé participants are recruited as a representative sample of Geneva noninstitutionalized residents aged 20 to 74. Between April and June 2020, former Bus Santé participants were recruited within the SEROCoV-POP study, along with their family members. Starting in December 2020, SEROCoV-POP participants were further invited to create an account on the dedicated digital platform Specchio-COVID19 [15], designed to follow-up adult participants of several sero-surveys carried out in the Canton of Geneva during the COVID-19 pandemic. The Specchio-COVID19 digital platform longitudinally collects from the same participants, information on self-reported symptoms and SARS-CoV-2 testing, infection risk and perception of exposure to risk, data on physical and mental health, and changes in working conditions.

The Corona Immunitas study [16] is a population-based, prospective cohort study purposely designed and conducted to assess the spread of the COVID-19 pandemic and its associated impacts, including on mental health. Participants were recruited after the first wave of the pandemic (in June 2020) and are followed-up for repeated serological testing, self-reported symptoms and assessments, including mental and physical health, psychological wellbeing, and lifestyle changes over time. Table A1 in the summarizes longitudinal data collection design across study sites and assessments.

### 2.2. Laboratory Analysis

Anti-SARS-CoV-2 immunoglobulin G (IgG) serology status was assessed in Geneva with a commercially available enzyme-linked immunosorbent assay (ELISA) (Euroimmun; Lübeck, Germany #EI 2606-9601 G), targeting the S1 domain of the spike protein. ELISA IgG ratio < 0.5 was indicative of seronegativity, while all sera with an IgG ratio of ≥0.5 were also tested with an immunofluorescence assay (IFA) and classified as either seropositive or seronegative according to the results from this test. In Ticino, in accordance with the Corona Immunitas study protocol [13], anti-SARS-CoV-2 IgG serology status was assessed using SenASTrIS (Sensitive Anti-SARS-CoV-2 Spike Trimer Immunoglobulin Serological) assay developed by the Centre Hospitalier Universitaire Vaudois (CHUV), the Swiss Federal Institute of Technology in Lausanne (EPFL) and the Swiss Vaccine Center [17].

### 2.3. Measurements and Procedures

We collected self-assessments of depressive symptoms using the 9-item Patient Health Questionnaire (PHQ-9), a 4-point depression module of PHQ (score 0 to 27) [18,19]. In Geneva, PHQ-9 data were collected between 2013 and 2019 within the Bus Santé study using paper-and-pencil questionnaires, which were completed at home and posted back via ordinary mail free of charges. In the Ticino site, PHQ-9 data were collected a few weeks before the blood collection appointment for the serological testing as part of an inclusion questionnaire using secured online questionnaires implemented in the Research Electronic Data Capture software (REDCap) [20], hosted at the Università della Svizzera Italiana. We computed a standardised continuous score of depression (with higher scores indicating higher depressive symptomatology) using a graded response model from the item response theory framework [21]. Cronbach α was equal to 0.84 in Geneva and 0.85 in Ticino.

Before the blood collection appointment, participants at both study sites received a link to online questionnaires covering socio-demographic information, health status, exposure to SARS-CoV-2-infected individuals, and self-reported COVID-19-compatible symptoms. Main participant characteristics included: age (categorised into aged 20–49, and aged 50–74), gender (women, men), smoking status (non-smoker, current smoker) and obesity (defined as body mass index ≥ 30 kg/m^2^, from self-reported body height and weight). Socio-demographic and health status indicators were comparable across study sites, and coded at baseline as follows: education (categorised into up to higher secondary/apprenticeship, and higher tertiary, work status (unemployed/retired, employed/student), and existing chronic conditions (none, any among hypertension, diabetes, cardiovascular disease, cancer, immunological syndrome, or respiratory syndrome). Presence of COVID-19 cases among close contacts or family members was assessed in both sites as a dichotomous variable with the following question: “How many people who live in your own household or with whom you are regularly in close contact, have tested positive for COVID-19 after taking a laboratory test?”. Participants were also asked to self-report whether they experienced any COVID-19-compatible symptoms not related to a known chronic condition since January 2020. To harmonize the lists of symptoms between study sites, we defined the following categories: systemic symptoms (i.e., fatigue, myalgia/arthralgia or loss of appetite); upper airways symptoms (i.e., either sneezing/rhinorrhoea or sore throat); gastro-intestinal symptoms (i.e., abdominal pain, nausea/vomiting or diarrhoea); fever and/or cough; dyspnoea; headache; and anosmia/dysgeusia [22] (Table A2). Within 8 subsequent weekly online follow-up surveys (during January and February 2021), participants were again asked to answer whether they experienced the same typologies of COVID-19 symptoms in the previous week.

### 2.4. Statistical Analysis

We modelled COVID-19 symptoms as binary dependent variables in separate Poisson regressions with robust standard errors [23,24] to estimate prevalence ratios (PR) and 95% confidence intervals (95% CI) by depression scores. At both baseline and follow-up assessments, analyses were run separately for each study site adjusting for age group, gender, education, work status, chronic diseases, obesity, smoking status, assuming invariance between follow-up assessments, and for presence of COVID-19 cases among close contacts or family members, and anti-SARS-CoV-2 IgG seropositivity. For Geneva, we further adjusted for the year of PHQ-9 completion (for data collected before the pandemic).

In further analyses, we added interaction terms between depression scores with serology status, gender and age group to test whether the association between depression and COVID-19 symptoms varied according to serological evidence for previous SARS-CoV-2 infection, gender or age, respectively. Likelihood ratio tests assessed significance and magnitude of interaction effects, comparing models with added interactions and models with no interactions [25]. For the rest of the analyses, we considered significance at *p* < 0.05 as standard. All analyses were performed using Stata 15 (StataCorp. 2015. Stata Statistical Software: Release 15. StataCorp LP, College Station, TX, USA).

## 3. Results

A total of 1157 adult participants from two large population-based observational studies (n = 576 Geneva Bus Santé and SEROCoV-POP studies; n = 581 Corona Immunitas Ticino study) were included in the analysis. Descriptive statistics of clinical and socio-demographic variables are reported in Table 1.

The flow-chart of recruited participants is reported in Figure 1. Between 2013 and 2019, we recruited 7785 participants in the Bus Santé study (mean participation rate was 50% over this period). Among these, 1618 (21%) took part to the psychological self-assessment returning their questionnaires via ordinary mail. Between April and June 2020, 716 among this smaller group of Bus Santé participants were recruited to SEROCoV-POP providing blood samples for laboratory analyses. To match samples across study sites in terms of age composition, we dropped SEROCoV-POP participants older than 64 years (n = 126) from the analytic sample. We also excluded participants with missing questionnaire data (n = 14), leaving a total of 576 (mean age 46 years; 58% females) for baseline analyses in Geneva. Among these, 365 (63% of the baseline sample; mean age 46 years; 61% females) also completed at least one weekly follow-up questionnaire with no missing information between January and February 2021, screening for COVID-19 symptoms in the previous week (Figure 1).

In July 2020, we recruited 1009 participants in the Corona Immunitas digital study in Ticino (participation rate was 27% accounting for all invitation letters sent to eligible subjects of the representative sample provided by the Federal Office of Statistics). Among these, 647 were randomly selected and provided blood samples for serological testing. We excluded 66 participants with missing data, obtaining an analytic sample of 581 (mean age 46 years; 57% females) in Ticino. Between January and February 2021, 536 (92% of the baseline sample; mean age 46 years; 58% females) completed at least one weekly follow-up questionnaire with no missing information, reporting whether they experienced any COVID-19 symptoms in the previous week (Figure 1).

In terms of frequencies of self-reported COVID-19-compatible symptoms across studies, we found that upper airways symptoms were the most reported at both baseline and follow-up in Geneva (baseline: n = 268, 47%; follow-up: n = 47, 13%) and Ticino (baseline: n = 162, 28%; follow-up: n = 105, 20%). The least frequently reported symptoms were anosmia/dysgeusia at both baseline and follow-up in Geneva (baseline: n = 39, 7%; follow-up: n = 6, 2%) and Ticino (baseline: n = 25, 4%; follow-up: n = 6, 1%) (Table A3). 

Figure 2 shows the results of Poisson regressions estimating likelihoods of declaring COVID-19 symptoms at baseline based on levels of depressive symptoms, controlling for sociodemographic factors (i.e., age, gender, education, work status), health status and behaviours (i.e., obesity, smoking, chronic diseases), and COVID-19 past exposure (i.e., presence of COVID-19 cases among close contacts or family members, anti-SARS-CoV-2 IgG serology status).

In Geneva, participants with higher levels of depressive symptoms were significantly more likely to report systemic (Prevalence Ratio (PR) = 1.15, 95% Confidence Intervals (95% CI): 1.03, 1.30, *p* = 0.017), upper airways (PR = 1.13, 95% CI: 1.03, 1.24, *p* = 0.010), gastro-intestinal symptoms (PR = 1.43, 95% CI: 1.19, 1.71, *p* < 0.001), and fever and/or cough (PR = 1.15, 95% CI: 1.02, 1.30, *p* = 0.023) (Table A4).

In Ticino, participants with higher levels of depressive symptoms were significantly more likely to report systemic (PR = 1.33, 95% CI: 1.15, 1.53, *p* < 0.001), upper airways (PR = 1.16, 95% CI: 1.01, 1.33, *p* = 0.035), gastro-intestinal symptoms (PR = 1.41, 95% CI: 1.06, 1.84, *p* = 0.016), fever and/or cough (PR = 1.19, 95% CI: 1.02, 1.39, *p* = 0.026), dyspnoea (PR = 1.39, 95% CI: 1.06, 1.82, *p* = 0.016), and headache (PR = 1.29, 95% CI: 0.85, 1.91, *p* = 0.007) (Table A5).

We found no significant interaction effects between depression and serology status, age group or gender (*p* > 0.05) (Table A6 and Table A7). Accordingly, in seronegative participants (n = 529 in Geneva, and n = 534 in Ticino) higher levels of depressive symptoms were associated with higher likelihoods of declaring COVID-19-compatible symptoms.

Figure A1 shows the results of Poisson regressions estimating likelihoods of reporting COVID-19 symptoms at follow-up. In both Geneva and Ticino, participants with higher levels of depressive symptoms were significantly more likely to report systemic symptoms (Geneva: PR = 1.72, 95% CI: 1.18, 2.51, *p* = 0.005; Ticino: PR = 1.29, 95% CI: 1.09, 1.54, *p* = 0.004) and dyspnoea (Geneva: PR = 3.34, 95% CI: 1.46, 3.62, *p* = 0.004; Ticino: PR = 1.63, 95% CI: 1.16, 2.31, *p* = 0.005) at follow-up (Table A8 and Table A9). 

Higher levels of depressive symptoms in Ticino were associated with upper airways symptoms (PR = 1.29, 95% CI: 1.10, 1.53, *p* = 0.002) and headache (PR = 1.59, 95% CI: 1.32, 1.92, *p* < 0.001) at follow-up (Table A9).

## 4. Discussion

The present study described the association between the severity of depressive symptoms and likelihoods of reporting COVID-19 symptoms in a large sample of adult participants based in Switzerland, a country that was severely hit by the COVID-19 pandemic. We found that higher self-reported levels of depressive symptoms were associated with higher likelihoods of reporting COVID-19 symptoms, including systemic symptoms, upper airways symptoms, gastro-intestinal symptoms, fever and/or cough, dyspnoea and headache. These associations did not vary by evidence of previous SARS-CoV-2 infection confirmed by serology.

Anosmia and dysgeusia were not associated with depressive symptomatology. This finding may indicate that the self-reported loss of smell and/or taste is indeed a specific, nearly pathognomonic COVID-19 symptom, which can be useful to identify people for further testing, although it occurs less frequently than other symptoms [26,27,28]. On the other hand, depressive symptoms were consistently associated with both long- and short-term recalls of dyspnoea and systemic symptoms (i.e., fatigue, myalgia/arthralgia or loss of appetite), that are listed among common manifestations of COVID-19 by the World Health Organization [29]. Depression is a frequent comorbidity in chronic respiratory and fatigue syndromes [30,31], and depression itself can be the cause of physical manifestations including fatigue, headaches, appetite loss, disseminated pain and dyspnoea, for example via comorbid anxiety [10,11]. Extensive exposure to alarmistic messages from authorities and the media during the pandemic months, including on the clinical manifestation and course of COVID-19, may have contributed to the association of depression with COVID-19 symptoms in our samples because this was independent of infection status.

### 4.1. Future Implications

Our findings nonetheless suggest that participants with higher depression levels may overreport certain categories of COVID-19-compatible symptoms. This can have important implications for screening and detection of possible or probable for SARS-CoV-2 infections, but also on the clinical management of depressive symptoms, particularly in relation to the implementation of indicated prevention strategies for lowering symptom levels and preventing their escalation to a formal psychiatric diagnosis [32,33].

Next, our results have important implications to inform public health strategies on testing, self-medication, quarantine, and self-isolation based on self-reported COVID-19 infection symptoms, which have also been used in observational studies as proxy of possible SARS-CoV-2 infections [28]. Self-reported symptomatology could be integrated with real-time tracking systems of self-reported potential COVID-19 symptoms such as smartphone-based apps, which may be better suited to collect predictive information for potential infection (including timing and frequency of symptoms occurrence) [34,35,36,37]. Nevertheless, accounting for responders’ mental health status could help refine algorithms triggering diagnostic work-ups, improving the appropriateness of testing, and contributing to averting the inflation of testing demand beyond public health infrastructure capacity.

Our findings are novel and are consistent, though somewhat difficult to compare with the results of previous studies. To the best of our knowledge, this is the first study specifically exploring the association between poor mental health and self-reporting of COVID-19 symptoms in a large sample of adults while further controlling for SARS-CoV-2 serology confirmed infection. Daniali and Flaten [38,39] observed how anxiety increased reports of COVID-like symptoms and beliefs of being infected by COVID-19. They interpreted these results as a nocebo effect, according to which negative expectations and health-related anxiety can lead to worsening of symptoms even in the absence of plausible biological mechanisms [40]. Other studies have reported an increase in general somatic symptoms, including fatigue, gastrointestinal symptoms, dyspnoea, headache, and nausea, following COVID-19-related anxiety [41,42,43,44]. Similarly, previous research tested whether anxiety and depression may confound chronic rhinosinusitis symptom reporting (including systemic symptoms, cough and fever), but did not find any significant association [45]. While evidence remains scarce [46], our results offer an insight into the potential confounding role of psychological symptoms for the detection of COVID-19 based on self-reported symptomatology.

### 4.2. Limitations and Strengths

This study was not without limitations. First, to keep data comparable across study sites, only a limited number of confounding factors could be included in the analyses. Residual confounding cannot be excluded. A more precise assessment of household income may have further contributed to explaining individual differences in reporting COVID-19 in association with depression. Moreover, a measure of health literacy could have contributed to disentangling the interaction between mental health perception and knowledge, psychological symptoms, and self-perception of SARS-CoV-2 infection, for example by acting as a moderator [47]. However, the effect of adjustments in our multivariate model was minimal. We tested prospective associations of past self-assessed depressive symptoms with long- and short-term recalls of more recent COVID-19 symptoms. We cannot exclude the possibility that associations between assessments taken at the same time vary. Moreover, mental health was assessed during and after the pandemic in Ticino and Geneva, respectively. Participation rates during the pandemic were higher than expected but participant selection in the study and the derivation of the analytic sample may have introduced some biases, though these were likely non-differential and may have contributed to shifting our estimates toward the null effect. For example, on the one hand, people experiencing or more prone to report COVID-19 symptoms may have been more likely to participate in the surveys, but on the other hand, depressive symptomatology, which includes lack of motivation and mild to severe psycho-motor retardation, may deter participation in research. Major strengths of our study include the internal replication of hypothesis testing in two large, representative, population-based samples, standardized assessments of mental health, robust and reliable data capturing, collection, and management tools in both sites, and the use of previously validated and highly reliable and valid serological testing for the detection of SARS-CoV-2 antibodies. The different timing of data collection between study sites further supports our finding that pre-pandemic depression levels were associated with COVID-19 symptom reporting in the medium-to-long term. Ongoing data collections at both sites within the Specchio-COVID19 and the Corona Immunitas Ticino study cohorts will extend the scope of the analyses presented here by including repeated assessments of serology status, self-reported COVID-19-compatible symptoms, as well as physical and mental health statuses.

## 5. Conclusions

This study is the first to show how depressive symptoms among adults are associated with a higher likelihood of recalling COVID-19-compatible symptoms, independently of serological evidence for previous SARS-CoV-2 infection. Individuals with higher degrees of depressive symptoms may benefit from public health advice to reduce the chances of misinterpreting symptoms comparable to COVID-19, for example, short-term preventative psychosocial interventions aimed at lowering psychological symptoms and improving coping strategies. The findings also have implications in the understanding of COVID-19 symptomatology based on self-reported information. In sum, mental health can have relevant public health implications for improving the use of self-reported COVID-19 symptoms to inform and prompt testing, self-medication, self-isolation and other measures aimed at mitigating and containing the spread of the COVID-19 pandemic.

## Figures and Tables

**Figure 1 ijerph-19-16696-f001:**
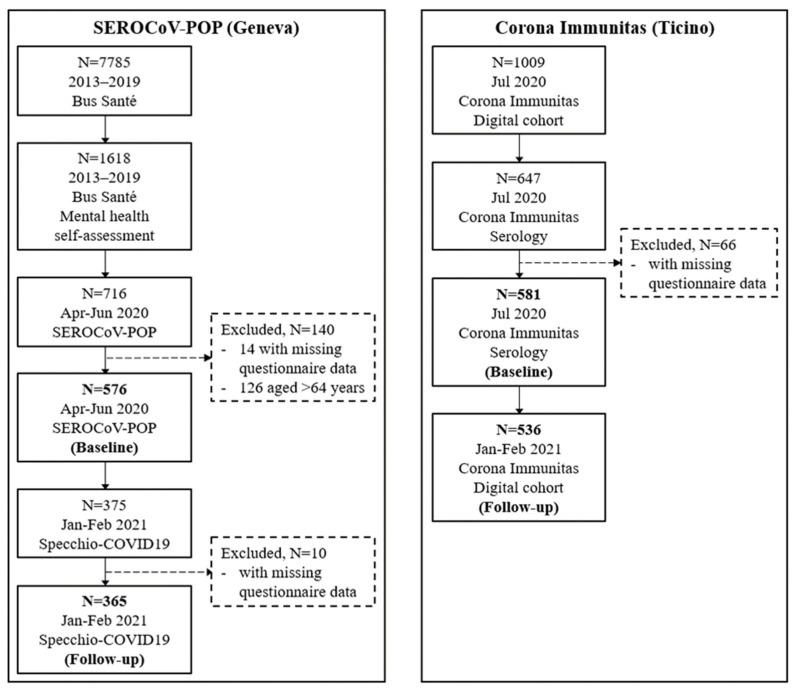
Flow chart for the selection of participants in Geneva and Ticino, Switzerland.

**Figure 2 ijerph-19-16696-f002:**
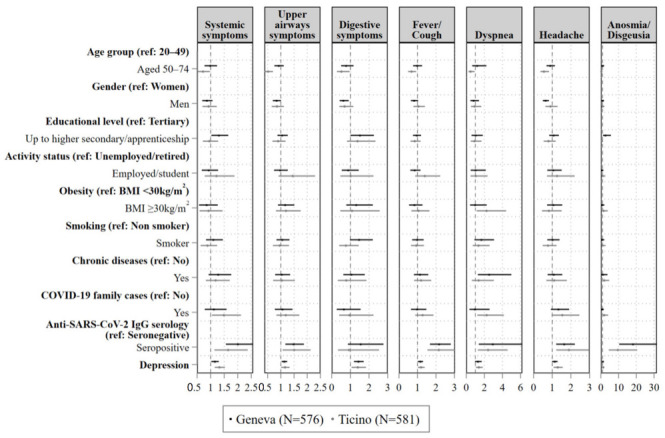
Results of Poisson regressions estimating likelihoods of reporting COVID-19 symptoms at baseline across studies. Note: values are prevalence ratios with 95% confidence intervals from Poisson regressions with robust standard errors, adjusted for all listed variables. For Geneva, analyses were further adjusted for the year of PHQ-9 completion.

**Table 1 ijerph-19-16696-t001:** Description of samples by study sites and assessments.

	Geneva		Ticino	
	Baseline (April–June 2020)	Follow-Up (February–January 2021)	Baseline (July 2020)	Follow-Up (February–January 2021)
N	576	365	581	536
Age, years, mean (SD)	46 (11)	46 (11)	46 (11)	46 (11)
Age group (years)				
20–49	333 (57.8)	212 (58.1)	327 (56.3)	293 (54.7)
50–64	243 (42.2)	153 (41.9)	254 (43.7)	243 (45.3)
Gender				
Female	333 (57.8)	222 (60.8)	328 (56.5)	310 (57.8)
Male	243 (42.2)	143 (39.2)	253 (43.5)	226 (42.2)
Educational level				
Up to higher secondary/apprenticeship	195 (33.9)	122 (33.4)	374 (64.4)	345 (64.4)
Tertiary	381 (66.2)	243 (66.6)	207 (35.6)	191 (35.6)
Work status				
Unemployed/retired	70 (12.2)	38 (10.4)	93 (16.0)	87 (16.2)
Employed/student	506 (87.8)	327 (89.6)	488 (84.0)	449 (83.8)
Obese (BMI ≥ 30 kg/m2)	47 (8.2)	24 (6.6)	62 (10.7)	57 (10.6)
Smoking				
No	487 (84.5)	320 (87.7)	453 (78.0)	421 (78.5)
Yes	89 (15.5)	45 (12.3)	128 (22.0)	115 (21.5)
Chronic diseases				
No	506 (87.9)	324 (88.8)	486 (83.7)	454 (84.7)
Yes	70 (12.1)	41 (11.2)	95 (16.3)	82 (15.3)
COVID-19 family cases a				
No	535 (92.9)	-	525 (90.4)	-
Yes	41 (7.1)	-	56 (9.6)	-
Anti-SARS-CoV-2 IgG seropositive a	47 (8.2)	-	47 (8.1)	-

Note: values are frequencies (percentages) unless stated otherwise. SD: standard deviation. BMI: Body mass index. Chronic diseases include hypertension, diabetes, cardiovascular disease, cancer, immunological syndrome, or respiratory syndrome. a—not assessed at follow-up.

## Data Availability

The data presented in this study are available on request from the corresponding author.

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
