# Peer review of "Associations between Depression and Self-Reported COVID-19 Symptoms among Adults: Results from Two Population-Based Seroprevalence Studies in Switzerland"

_ijerph, 2022, doi:10.3390/ijerph192416696_

Round 1

Reviewer 1 Report

First, i would like to compliment the authors for the time and effort, put into the protocol writing, design and manuscript writing.

The authors studied the association between depression using a validated psychometric scale, and COVID-19 symptomatology among adults, derived from two seroprevalence studies in Switzerland.

Overall, the authors used an adequate scientific language to communicate clearly the main findings from their study; the epidemiological design is well designed and the decision of using a poisson regression is a nice idea for the data in this study.

Therefore, after carefully reading the manuscript i recommend the manuscript for publication.

Author Response

Thanks for your appreciation of our work. 

Reviewer 2 Report

Thank you so much for allowing me to review this paper.

This article reveals the positive associations between depressive symptoms and COVID-19 36 compatible symptoms, independently of seropositivity. The self-reported infection symptoms may influence testing, self-medication, and containment measures, such as quarantine and isolation, with relevant public health implications.

I quite appreciated the author's effort from this perspective; I immensely enjoyed this work; however, I found some suggestions for this manuscript for minor revision.

Q1. Further extended some implications of the outcome from the abstract.

Q2. Suggested explaining more correlation between depression and self-reported COVID-19 2 symptoms. 

Q3. Why specially choose the target of adults instead of teens or older adults? Any comparison difference? If yes, please explain in the introduction section and the literature review support. 

Q4. Please consist the wording  COVID-19 pandemic or  COVID-19 epidemic. 

Q5. There are some empty columns in Annex Table S1; enter "N/A" for them.

Q6. Line 169 suggested authors explain "significance was considered for 169 p<0.05", how and why significance. 

Q7. Use Male and Female instead of Women and Men, as suggested in table 1. 

Q8. In Figure 1, what does missing data or an incomplete questionnaire mean?

Q9.Suggest adding the "Future implication"  and the "limitation" section.

Q10. For the citation that develops more about COV-19 and patient situation, I suggest adding the reference paper https://doi.org/10.3389/fmed.2021.666973.

Author Response

Thanks for your comments. Please, find our responses in the attachment.

Reviewer 3 Report

Manuscript ID: ijerph-2067854

This is an interesting paper about association between the severity of depressive symptoms and likelihood of reporting COVID-19 symptoms in a sample of adult participants in Switzerland.

In order to improve the overview of the results, some changes can be considered:

Line 97 - The Annex Table S1 should be included in Additional file (Appendix).

Line 101, 102 - Protocol numbers approved by the Ethics Committees are included in the section Institutional Review Board Statement (Line 408). It is not necessary to repeat them in the Materials of Methods.

Line 151 - The Annex Table S2 should also be included in Additional file (Appendix).

Line 177 - In Table 1. percentage results should be presented as whole numbers (same as in the text), without a decimal point. The same applies to Annex Table S3 (Line 216). Age, years, mean (SD) in Table 1. should be presented more simply [e.g. “45.75 (10.95)” as “46 (±10.9)”].

Line 197 - The flow chart for the selection of participants in Geneva and Ticino is explained in detail in the text of the Results. Figure 1 is just an unnecessary repetition of data.

Line 216 - In Annex Table S3. percentage results should be presented as whole numbers (same as in the text).

Line 235 - In Annex Table S4., Table is too large. The text in the Table can be described as follows:

Age group (ref: Aged 20-49)

Aged 50-64 ……

The meaning of each data separately [0.98 (0.77, 1.24)] should be described in the Table.

The meaning of  * and ** should also be described in the Table.

The same applies to Annex Table S5, S8 and S9. The meaning of  *** should also be described in the Table.

Line 251 – In Annex table S6. show statistically significant data more clearly.

Line 280. - Is it necessary to present the same results in two different ways (Annex Figure S1. vs. Annex Table S8. and S9.) (Figure 2. vs. Annex Table S4. and S5.)?

Author Response

(The authors gave the same response as above.)
